# A visual opsin from jellyfish enables precise temporal control of G protein signalling

Michiel van Wyk [1,2] ✉ & Sonja Kleinlogel[1,2,3] ✉

Phototransduction is mediated by distinct types of G protein cascades in different animal taxa: bilateral invertebrates typically utilise the Gαq pathway whereas vertebrates typically utilise the Gαt(i/o) pathway. By contrast, photoreceptors in jellyfish (Cnidaria) utilise the Gαs intracellular pathway, similar to olfactory transduction in mammals[1]. How this habitually slow pathway has adapted to support dynamic vision in jellyfish remains unknown. Here we study a light-sensing protein (rhodopsin) from the box jellyfish *Carybdea rastonii* and uncover a mechanism that dramatically speeds up phototransduction: an uninterrupted G protein-coupled receptor – G protein complex. Unlike known G protein-coupled receptors (GPCRs), this rhodopsin constitutively binds a single downstream Gαs partner to enable G-protein activation and inactivation within tens of milliseconds. We use this GPCR in a viral gene therapy to restore light responses in blind mice.

Box jellyfish have camera-type eyes, with a cornea, lens and retina[2]. They use visual information in a range of behaviours, including object avoidance and active hunting. The G-protein coupled rhodopsins of jellyfish and other members of the cnidarian phylum signal via the Gαs pathway, similar to our sense of smell, whilst vision in other animals is typically signalled via either the Gαq pathway (bilateral invertebrates) or the Gαt(i/o) pathway (vertebrates)[1]. How box jellyfish achieve fast vision through this unconventional pathway remains a mystery. Here we show that, unlike canonical G-protein coupled receptors (GPCRs), an opsin-based pigment from the box jellyfish *Carybdea rastonii*, JellyOp, constitutively binds a single Gαs protein to create a direct link between light detection and G-protein signalling. We introduce this unconventional opsin to mammalian cells and use light stimuli of different colours to activate and inactivate G-protein signalling within milliseconds. Indeed, since JellyOp controls the activity state of its bound G-protein a in an allosteric manner, responses are independent of G-protein dynamics and cyclic nucleotide exchange, which normally introduce substantial response delays[3,4].

In this work, we demonstrate the unprecedented speed and fidelity at which JellyOp/Gαs controls the release of Gβγ subunits and capitalise on this property in several ways. (1) We engineer a JellyOp-Gαs fusion protein with strong and selective coupling to the Gβγ

pathway—the first optogenetic tool of its kind. (2) We use JellyOp in a viral gene therapy to drive Gβγ-signalling in retinal neurons and restore fast light responses in the retinas of blind mice.

## Results

Since Gαs signalling is atypical in visual transduction, we first tested if JellyOp, which was previously shown to drive Gαs signalling, also couples to G-proteins in the Gαi/o family[1]. To do this, we introduced JellyOp to HEK293 cells (Fig. 1a) and monitored the light-triggered rise in intracellular cAMP with or without pre-incubation with pertussis toxin (PTX), a potent inhibitor of the Gαi/o class of G-proteins[5]. PTX caused a small but significant increase in the cAMP signal. Therefore, despite the clear preference for Gαs shown previously, JellyOp also couples to Gαi/o (Fig. 1b). Further to these findings, we tested if JellyOp activates GIRK channels, which are classically gated by Gβγ subunits released from G-proteins in the Gαi/o family[6]. JellyOp consistently activated GIRK currents in different GIRK expression systems—both in HEK293-GIRK cells that stably express the GIRK1/2 channel (Fig. 1c) and in a cardiomyocyte cell line (HL-1) that endogenously expresses GIRK1/4 channels (Fig. 1d). Unlike GIRK responses signalled through other GPCRs, the JellyOp-triggered GIRK current was biphasic: a remarkably fast onset with a time constant, TauON, of $34 \pm 1$ ms (Fig. 1e) was

[1]Department of Biomedical Research, University of Bern, Bern, Switzerland. [2]Institute of Physiology, University of Bern, Bern, Switzerland. [3]Present address: Roche Pharma and Early Development, Neuroscience and Rare Diseases, Roche Innovation Center, F. Hoffmann-La Roche Ltd, Basel, Switzerland. ✉e-mail: michiel.vanwyk@unibe.ch; sonja.kleinlogel@roche.com

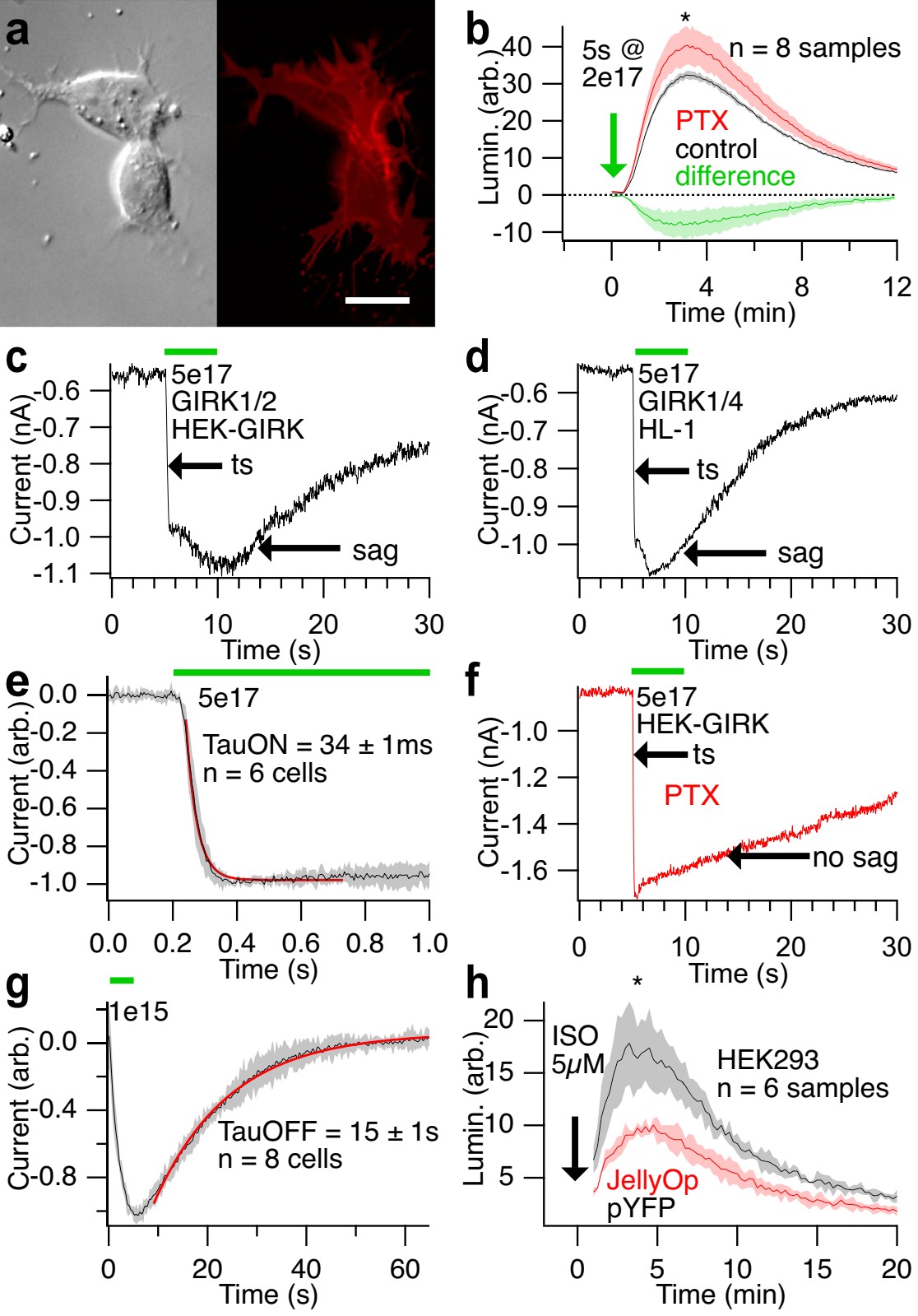

overlaid by a smaller, slower component (sag) with kinetics more characteristic of GIRK currents triggered by Gαi/o-coupled rhodopsins, such as human middle-wave opsin (OPN1MW; see below). Preincubation with PTX selectively blocked the small sag in the GIRK response to isolate a robust, ultrafast GIRK current signalled through Gαs (Fig. 1f). These findings show that GPCR-to-GIRK coupling can be efficiently driven by Gβγ without a specific requirement for Gαi/o

subunits[7,8]. Indeed, JellyOp signalled via Gαs to activate GIRK currents more than thirty times faster than any known Gαi/o-coupled GPCR[9]. Inactivation of the GIRK current was more orthodox, with a time constant, TauOFF, of ~15.3 ± 0.1 s (Fig. 1g).

The extremely fast onset and early peak of the JellyOp GIRK response suggests that there is a pre-association between JellyOp and Gαs, already in the inactive (dark) state of the opsin-based pigment.

**Fig. 1 | JellyOp activates robust ultrafast GIRK currents via Gαs. a** Differential interference contrast and epi-fluorescence micrographs of the same HEK293 cells transfected with a JellyOp-mKate fusion protein demonstrate highly efficient membrane targeting of JellyOp. Scale bar is 20 μm. **b** cAMP bioluminescent plate reader assay on JellyOp transfected HEK293 cells performed with (red) and without (black) PTX shows a preference for Gαs but significant coupling to Gαi/o ($p = 0.02$; one-sided $t$-test; error bands show mean ± SD). **c,d** JellyOp efficiently activates GIRK channels with a transient (ts) response onset followed by a small slower sag current. Similar GIRK responses were recorded in stable GIRK1/2 expressing HEK-GIRK cells (**c**) and in endogenously GIRK1/4 expressing HL-1 cells (**d**). **e** The transient response component has an ultrafast onset. **f** The sag current is abolished by pre-treatment with PTX, indicative of being mediated via Gαi/o (error bands show mean ± SD). **g** The JellyOp triggered GIRK current recover in the dark with kinetics typical for GPRC rhodopsins (error bands show mean ± SD). **h** The cAMP response of endogenous Gαs-coupled β2-adrenoceptors in HEK293 cells stimulated with iso-proterenol (ISO) is significantly smaller in cells transfected with JellyOp (red; no illumination; 1 μM 9-cis-retinal; $p = 0.027$ at peak; one-sided $t$-test; error bands show mean ± SD) compared to control cells transfected with YFP (black). The residual ISO-induced rise in cAMP seen in the red trace likely arises from cells not expressing JellyOp (transfection efficacy ~50%). Light stimulation (500 nm) is indicated as green bars and light intensities are given in figure panels (photons/cm²/s). Source data are provided as a Source Data file.

To explore this more closely, we used a competition assay where JellyOp and β2-adrenergic receptors compete for the same intracellular complement of Gαs[10]. We found that the rise in cAMP signalled by β2-adrenergic receptors endogenously expressed in HEK293 cells (induced by 5μM isoproterenol) was significantly reduced solely by transient expression of JellyOp without light activation (Fig. 1h)[11]. This suggests that dark-state JellyOp scavenges the available intracellular pool of Gαs from activated β2-adrenergic receptors. To support these findings and demonstrate formation of a dark-state JellyOp/Gαs complex, we co-transfected HEK293 cells with a JellyOp-IRES-TurboFP635 plasmid and a NES-venus-MiniGαs plasmid in the absence of retinal[12]. We then imaged the same cells before and after a 10 min incubation period with 1μM 9-cis-retinal in the dark (Fig. 2a–c). The results were conclusive. When the JellyOp apoprotein was reconstituted to the dark-state with 9-cis-retinal, venus-MiniGαs proteins were recruited to the cell membrane only in cells that also expressed JellyOp (Fig. 2e,f). Interestingly, once recruited to the cell membrane, venus-MiniGαs remained at the cell membrane despite extensive light stimulation (Fig. 2d). These findings not only confirm a pre-bound JellyOp/Gαs complex, but one that remains uninterrupted during light-activation and signalling.

We next measured the light sensitivity of the JellyOp GIRK response. The light intensity that evoked half the maximal GIRK response ($P_{50}$) was $1.05 \times 10^{13}$ photons/cm²/s for a 5 s stimulus, equivalent to a bright daylight intensity (Fig. 3a, b)[13]. We also confirmed a spectral sensitivity peak at ~490 nm (Supplementary Fig. 1a,b)[1]. The fast kinetics of the GIRK assay allowed us to revisit basic properties of the JellyOp photocycle. We show that JellyOp can be repetitively activated at ~1 min intervals without signal attenuation; such a demonstration was not possible with the relatively slow cAMP assays used previously (Supplementary Fig. 1c, d)[5]. Indeed, these response dynamics were only limited by the speed at which the GIRK response recovered. Since the active state of JellyOp was previously shown to have a blue-shifted absorbance compared to the dark state (albeit with no evidence for bi-stability), we re-examined potential bi-stability by activating cells with 500 nm green light and subsequently illuminating the active state with 405 nm violet light, corresponding to the wavelength where the published dark- and light-absorbance spectra were most separated[1]. We found that the JellyOp-triggered GIRK current is rapidly turned off by high-intensity violet light (Fig. 3c–e). We were not able to reactivate GIRK currents for several minutes after inactivation (tested ≤ 20 min). Therefore, we conclude that JellyOp is not a classic switchable bi-stable opsin but, rather, that it can be inactivated by violet light.

Once again, speed was the surprising factor. Violet light inactivated the JellyOp GIRK response much faster than GPCR signalling can normally be turned off (Fig. 3f; TauOFF = 252 ± 5 ms). For comparison, activated Gαs and Gβγ subunits re-associate to their inactive conformation with time constants ranging from ~15 to 40 s when canonical Gαs-coupled receptors turn off–in the same HEK293 expression system[14]. Much faster inactivation of the JellyOp GIRK response implies a single JellyOp/Gαs functional unit that can be directly activated and inactivated by light to rapidly release or bind Gβγ. Since we were not

able to separate individual time constants for Gαs inactivation and Gβγ scavenging from GIRK channels, we conclude that JellyOp inactivates its bound Gαs partner at a time constant of at least 252 ± 5 ms and that this inactivation could be much faster. Hence, these data complement our Mini-Gαs results (Fig. 2) and again infer an uninterrupted active-state JellyOp/Gαs complex.

Next, we mutated JellyOp with the aim to weaken its affinity for Gαs and simulate signalling through canonical GPCRs. More than 20 amino acid residues of bovine rhodopsin were previously identified as potential binding partners to Gαt[15,16]. We identified corresponding residues in JellyOp by homology modelling to bovine rhodopsin and identified K72 (numbered according to bovine rhodopsin) in the first intracellular loop of JellyOp as a target for mutagenesis. We selected this residue for two reasons: (1) Lysine is an unconventional residue at this GPCR-Gα interaction site, which is typically populated by a hydrophobic or uncharged amino acid in other Group A GPCRs. (2) In-silico mutagenesis of rhodopsin/MiniGαo (6FUF) suggests a unique hydrogen bond between K72 of JellyOp and Q350 of Gαs (Fig. 4a; a covalent interaction was ruled out, Supplementary Fig. 2)[16]. To weaken G-protein affinity without disrupting Gαs specificity, we replaced this lysine residue by a threonine. Threonine is often found in this location in other Gαs-coupled GPCRs, including the β1-adrenergic receptor which couples exclusively to Gαs (Fig. 4a)[17]. As anticipated, the JellyOp(K72T) mutant activated GIRK currents with slower kinetics compared to WT JellyOp and typical for Gαi/o-coupled rhodopsins, inferring the absence of a rigid pre-coupled JellyOp(K72T)/Gαs complex (Fig. 4b,c). To confirm that JellyOp(K72T) and Gαs do not form a pre-bound complex, we co-transfected HEK293 cells with JellyOp(K72T) and NES-venus-MiniGαs and show that JellyOp(K72T) does not recruit NES-venus-MiniGαs to the cell membrane in the presence of 9-cis-retinal (Supplementary Fig. 3). Since JellyOp(K72T) is not constitutively bound to a single Gαs subunit, we expected that photo-stimulation would activate multiple G-proteins to produce more Gβγ subunits and activate more GIRK channels. However, the amplitude of Gαs-mediated GIRK currents (after PTX treatment) did not increase in the K72T mutant (Fig. 4d). The most parsimonious explanation seems to be that both, WT JellyOp and JellyOp(K72T), when transiently overexpressed in a heterologous system such as HEK293-GIRK cells, activate nearly the entire complement of Gαs proteins within the cell. In support of this hypothesis, a JellyOp-Gαs fusion protein triggered significantly larger GIRK currents compared to WT JellyOp (Supplementary Fig. 4a–c). Astoundingly, despite strong GIRK activation, the same fusion protein did not couple to adenylyl cyclase (AC) and therefore present a useful tool for selective manipulation of the Gβγ pathway (Supplementary Fig. 4d).

Inactivating the JellyOp(K72T) mutant with violet light resulted in a clear double-exponential cessation of the GIRK current. The fast component imitates inactivation of WT JellyOp (TauOFF1 = 281 ± 10 ms), while the slower component was similar to the TauOFF of G-proteins activated by other Gαs-coupled GPCRs in HEK293 cells (TauOFF2 = 8.4 ± 0.6 s; n = 4 cells; Fig. 4e)[14]. These data indicate that, although JellyOp(K72T) does not pre-bind Gαs in the dark state, a fraction of active JellyOp(K72T) molecules are bound to activated Gαs.

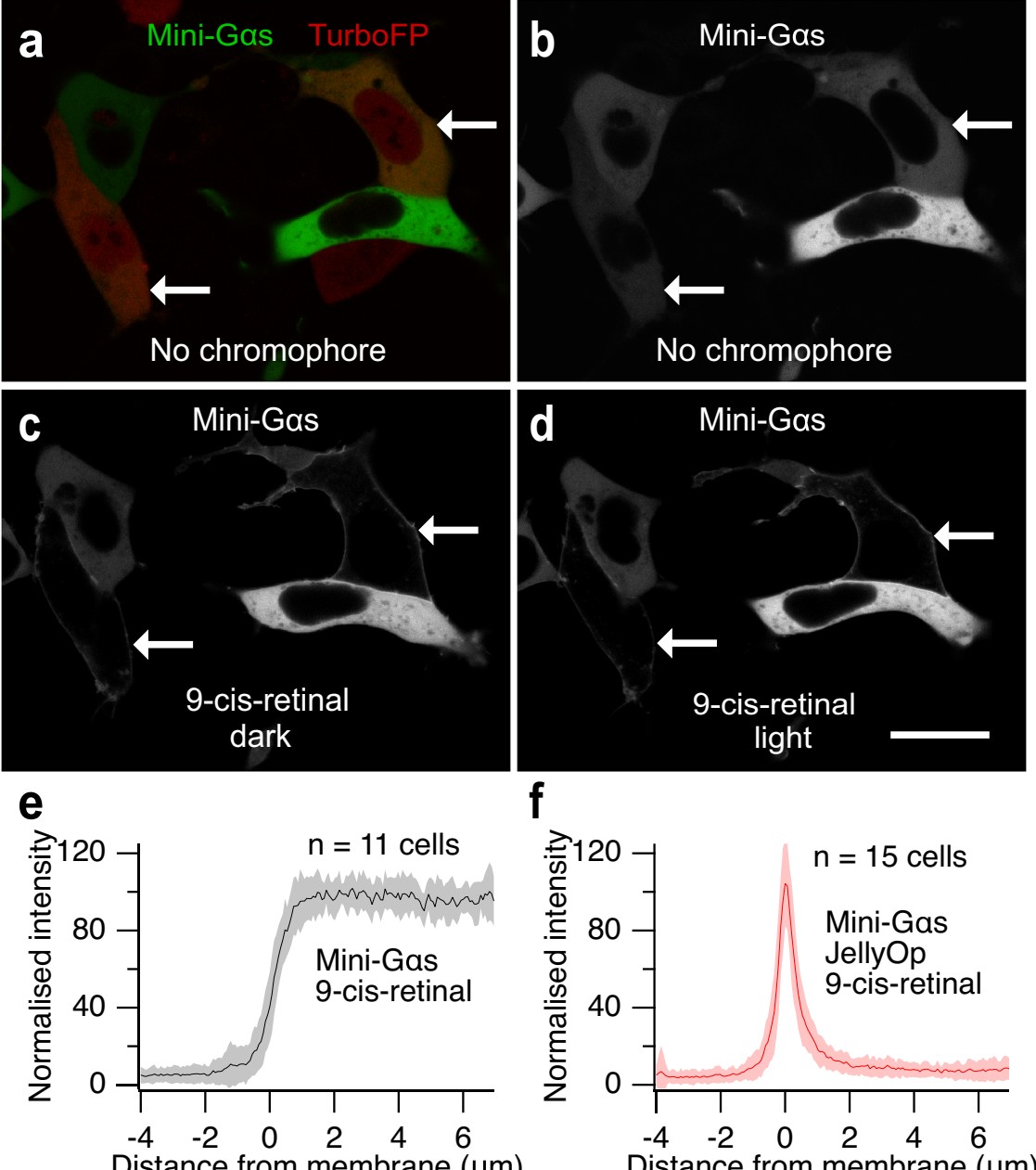

**Fig. 2 | JellyOp recruits NES-venus-MiniGαs to the cell membrane. a** A field of HEK293 cells transiently transfected with NES-venus-MiniGαs (green) and JellyOp-IRES-TurboFP635 (red), with two cells that express both plasmids (arrows). **b** The green channel from (**a**) shows that NES-venus-MiniGαs is in the cytoplasm in the absence of a chromophore. **c** After adding 9-cis-retinal in the dark (1 µM for 10 min), NES-venus-MiniGαs is recruited to the cell membrane, only in the cells that express

JellyOp (arrows). **d** NES-venus-MiniGαs remains at the cell membrane during subsequent light exposure (500 nm; $5 \times 10^{13}$ photons/cm$^2$/s; 10 min). Scale bar is 20 µm. **e,f** Normalised fluorescence intensity of NES-venus-MiniGαs measured across the cell membrane of cells without (**e**) and with (**f**) reconstituted JellyOp (error bands show mean ± SD). Source data are provided as a Source Data file.

We therefore conclude that JellyOp not only holds on to its pre-bound Gαs partner during signalling, but it develops an even higher affinity for Gαs in the active conformation. The location of the K72T mutation (at a demonstrated G-protein association site) suggests that Gαs most likely binds at a conventional locus on the intracellular surface of JellyOp in both the active and inactive states[16].

Once activated, Gαs subunits leave the plasma membrane[18]. If JellyOp keeps active Gαs on the plasma membrane, in proximity to AC, it might be particularly efficient at raising intracellular cAMP. That is what we observed. WT JellyOp triggered a significantly larger rise in intracellular cAMP compared to JellyOp(K72T), which again supports the idea of an active-state JellyOp/Gαs complex (Fig. 4f). Indeed, since

Gβγ-gated channels were never identified in the jellyfish eye, the native effect of constitutive coupling between JellyOp and Gαs may relate to an enhanced Gαs signal.

However, despite strong coupling to AC, it is the speed and efficacy at which JellyOp drives Gβγ release from Gαs that is truly exceptional. Given this property, we exploited JellyOp as an optogenetic tool to drive Gβγ signalling in retinal ON-bipolar cells (OnBCs; second-order retinal neurons) to ultimately restore behavioural vision in blind mice. We used an in vivo AAV-mediated gene therapy to ectopically express JellyOp in OnBCs, where excitability is naturally controlled through Gβγ-gated TRPM1 non-selective cation channels[19–21]. By treating blind mice that suffer from photoreceptor

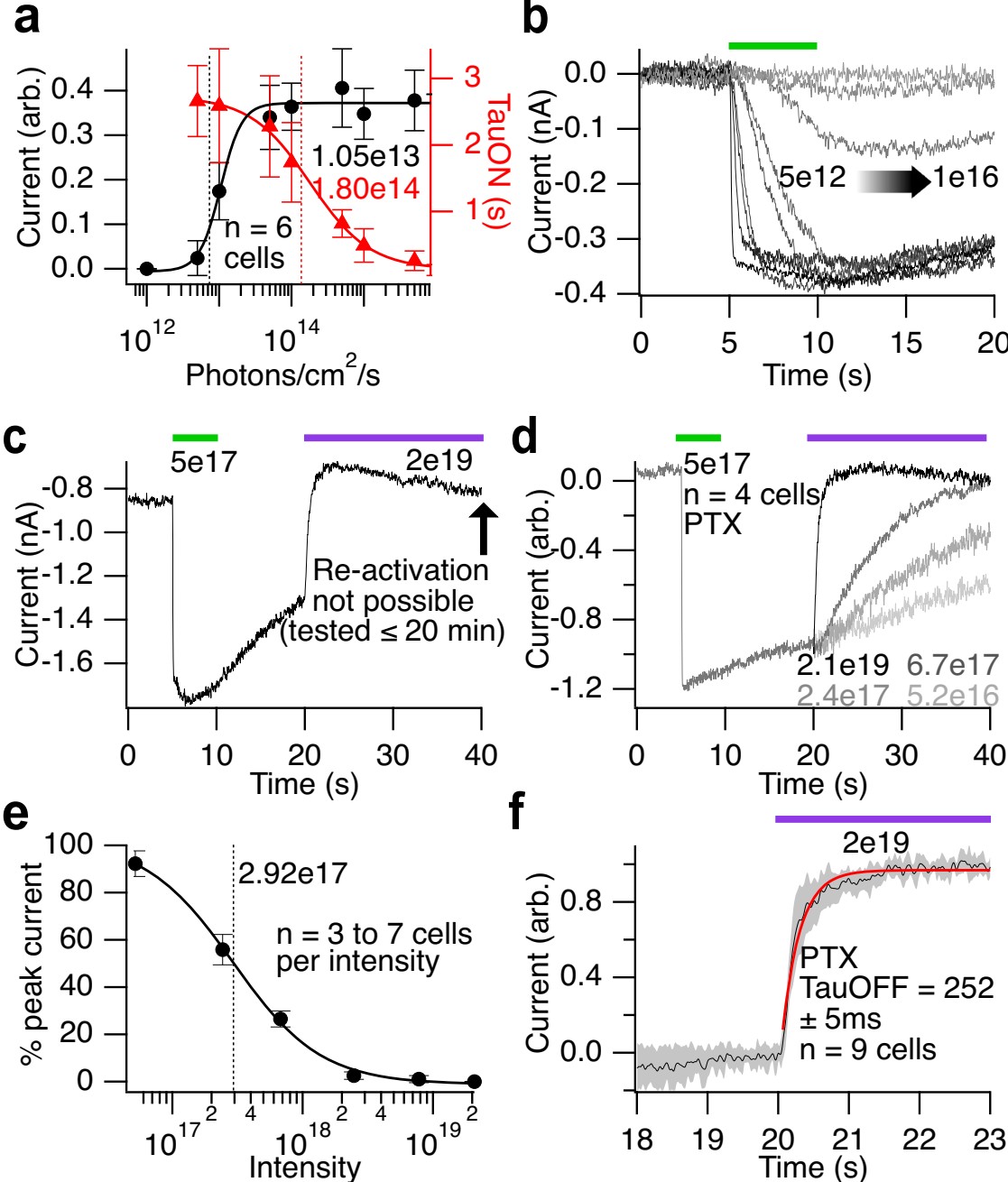

**Fig. 3 | Light sensitivity of the JellyOp-triggered GIRK current. a** The light intensity required for 50% maximum GIRK current activation by green light ($P_{50}$) was $1 \times 10^{13}$ photons/cm²/s (black trace; 5 s stimulation at 500 nm; show is mean ± SD). The speed of GIRK activation increased at higher intensities (red; $TauON_{50} = 1.8 \times 10^{14}$ photons/cm²/s; show is mean ± SD). **b** Example traces used to generate the light intensity curves in (**a**). **c,d** JellyOp-triggered GIRK currents are inactivated within milliseconds by high-intensity violet (405 nm) light, shown without (**c**) and with PTX pre-treatment (**d**). **e** Light intensity response curve showing the percentage of GIRK current inactivated by a violet light stimulus presented for 5 s (405 nm; $P_{50}$ (Hill fit) = $2.9 \times 10^{17}$ photons/cm²/s; the number of cells used for each data point (from the lowest to the highest violet intensity) is $n = 3, 6, 3, 4, 3$ and 7; show is mean ± SD). **f** The time constant of GIRK inactivation at the highest violet light intensity tested was 250 ± 5 ms. Error bands show mean ± SD. Light stimulation is shown as bars (green = 500 nm; violet = 405 nm) with intensities indicated on the figure panels (photons/cm²/s). Source data are provided as a Source Data file.

degeneration (*rd1* mouse line), we hoped to restore fast light responses by rendering the visual signalling cascade in the OnBCs directly light sensitive through JellyOp[22]. Since TRPM1 in OnBCs is naturally gated via a Gαo-coupled GPCR (mGluR6; senses glutamate released from photoreceptor cells), we first confirmed that OnBCs express sufficient levels of Gαs by performing OnBC-specific RT-qPCR Gαs quantification[23,24]. Although OnBCs contained about eleven times more Gαo mRNA than Gαs mRNA (Fig. 5a), we anticipated

this level of Gαs to suffice, given the demonstrated high affinity of JellyOp for Gαs.

OnBC-targeted expression of JellyOp-IRES-TurboFP635 was robust (Fig. 5b). The light-responses of isolated transduced OnBCs—specifically rod-BCs—had a fast onset comparable in kinetics to the transient GIRK currents observed in HEK293-GIRK and HL-1 cells and consistent with fast TRPM1 gating by released Gβγ from JellyOp/Gαs (Fig. 5c; $n = 4$ cells). These light responses in rod-BCs were significantly

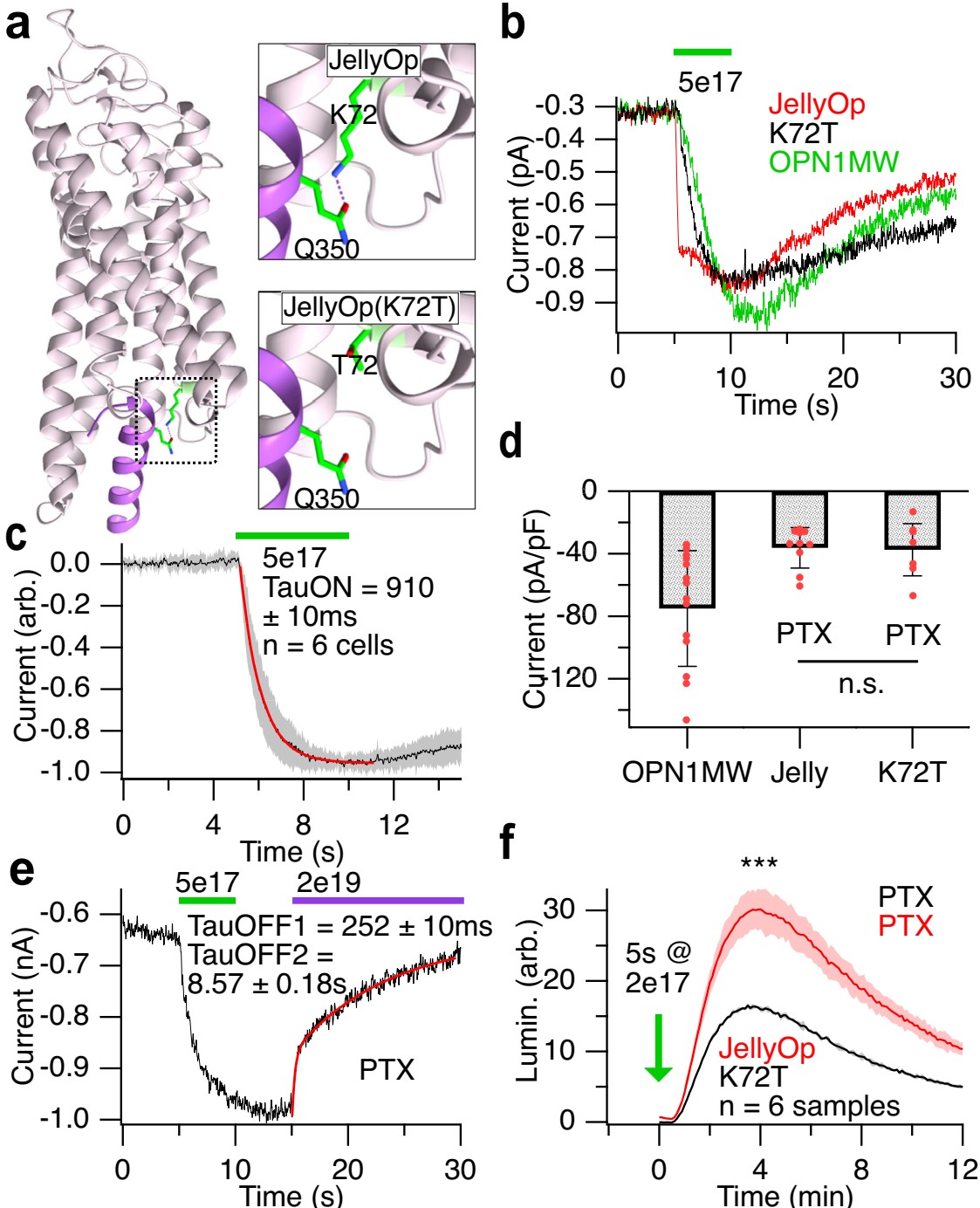

**Fig. 4 | Weakening the affinity of JellyOp for Gαs by introducing the K72T point mutation confirms an uninterrupted JellyOp/Gαs complex. a** The crystal structure of rhodopsin/Mini-Gαo (6FUF) with residues corresponding to L72 of bovine rhodopsin and G350 of Gαo mutated to the corresponding residues in JellyOp and Gαs (using ChimeraX) predicts a hydrogen bond interaction (low mag and top insert), which is no longer present when K72 is mutated to T (bottom insert). **b** GIRK currents activated by the K72T mutant have a delayed response onset, similar to that of OPN1MW. **c** The onset of JellyOp(K72T)-triggered GIRK currents have a significantly slower time constant than currents triggered by WT JellyOp ($p < 0.001$; error bands show mean ± SD; see Fig. 1e). **d** Despite losing its transient response component, the K72T mutant still triggers GIRK responses in the

presence of PTX of a similar amplitude compared to WT JellyOp, indicating that a similar amount of G-proteins are activated ($p = 0.8$; two-sided $t$-test; $n = 14$ cells for OPN1MW, 10 cells for JellyOp and 7 cells for JellyOp(K72T); shown is mean ± SD). **e** Example trace showing that bright violet light inactivates the JellyOp(K72T) GIRK current with a double exponential decay. Shown variance of fit coefficients is SD. **f** Despite similar GIRK current amplitudes observed in panel d, the rise in cAMP stimulated by JellyOp(K72T) was significantly reduced ($p < 0.001$; two-sided $t$-test; error bands show mean ± SD). Stimuli shown in green is 500 nm and in violet is 405 nm with intensities indicated on panels (photons/cm²/s). Source data are provided as a Source Data file.

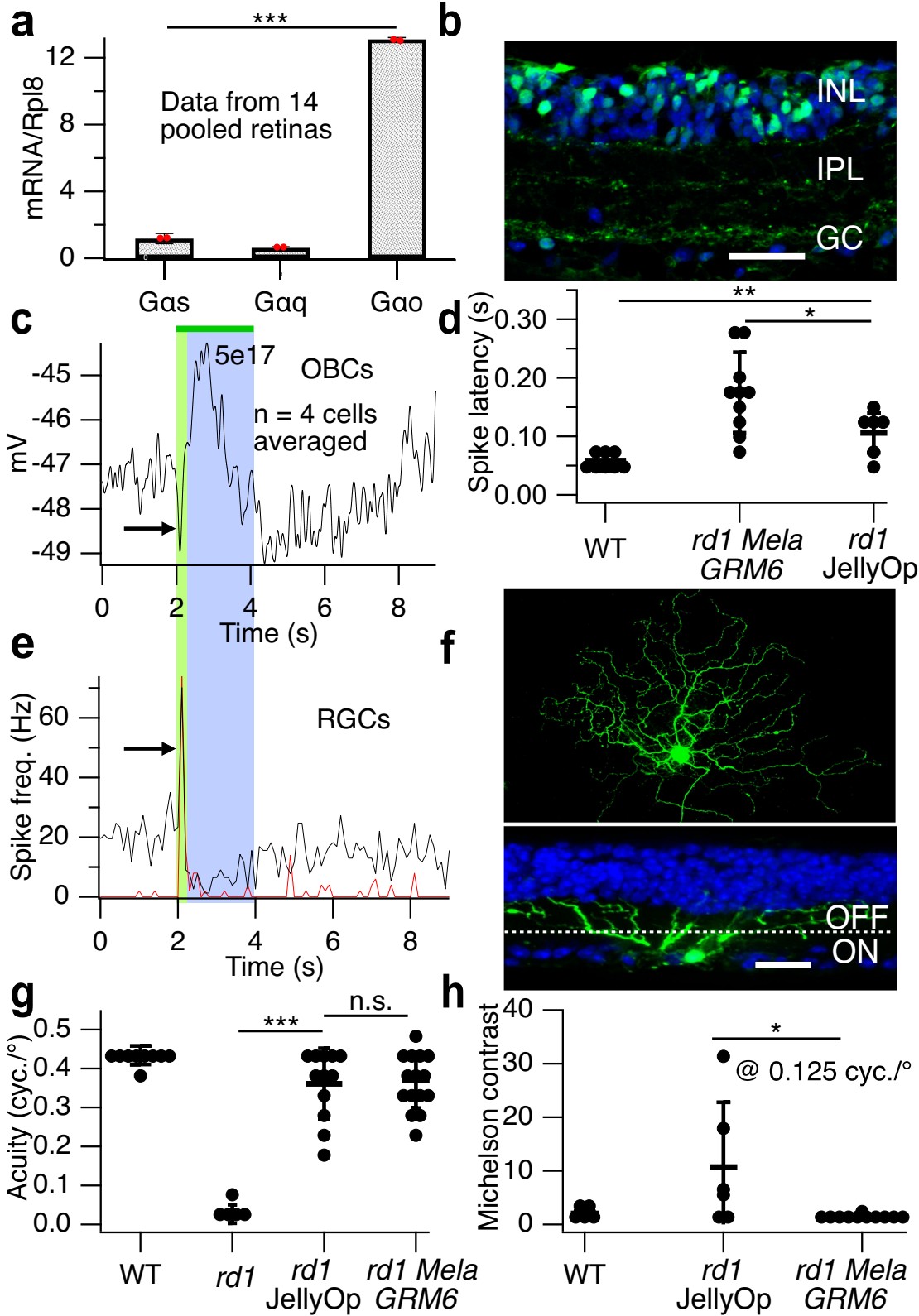

faster than those elicited by the designer Mela(CTmGluR6) optogenetic tool ($p < 0.001$, timepoint of peak hyperpolarization after start of light stimulus; Supplementary Fig. 5)[25]. This confirms the unparalleled speed, efficacy and sensitivity of Gβγ signalling through the JellyOp/ Gαs pair.

The initial hyperpolarization in the rod-BC light response was immediately followed by a broader depolarisation (Fig. 5c). It is known

that both, Gβγ and Gαo can bind to TRPM1 and cooperate to close the channel[26]. One possible explanation is that Gβγ released from JellyOp/ Gαs may initially close TRPM1 but also scavenge free Gαo subunits from TRPM1 to have a secondary opposing effect. Subsequent extracellular recordings from the retinal output neurons, the retinal ganglion cells (RGCs; $n = 18$), showed that the rapid hyperpolarization of OnBCs causes a rapid temporally aligned depolarisation of some RGCs

**Fig. 5 | Harnessing the efficient Gβγ signal of JellyOp to restore visual function in blind *rd1* mice. a** RT-qPCR from OnBCs of *rd1* retinas show that Gαs is expressed, albeit at a significantly lower level compared to Gαo ($p < 0.001$; two-sided *t*-test; qPCR template was pooled OnBC cDNA from $n = 14$ retinas; indicated is the SD of $n = 2$ technical replicates). **b** AAV(7m8)-JellyOp-IRES-TurboFP635 efficiently transduces OnBCs residing in the inner nuclear layer (INL) of the *rd1* retina after intravitreal injection as indicated by staining for the TurboFP635 reporter (green). Nuclei are labelled by DAPI (blue). The INL, inner plexiform layer (IPL) and ganglion cell (GC) layer are indicated. **c** Averaged voltage responses of isolated rod-BCs (single traces from $n = 4$ cells) that express the TurboFP635 reporter show a biphasic response that start with a fast transient hyperpolarization (arrow). **d** The onset of spikes in RGCs without spontaneous activity was significantly faster in JellyOp-treated *rd1* mice compared to Mela(CTmGluR6) treated *rd1* mice ($p = 0.019$; two-sided *t*-test) but still slower compared to healthy mice ($P = 0.002$; two-sided *t*-test; $n = 8$ WT mice, 10 *rd1* Mela(CTmGluR6) mice and 6 *rd1* JellyOp mice; shown is mean ± SD). **e** A significant fraction of RGCs in treated *rd1* retinas (8 out of 18 recorded light responsive cells) also responded with a transient increase in firing at

the onset of the light stimulus to mirror the rapid light responses recorded in OnBCs. Fast transient-ON responses in (**c** and **e**) are highlighted in green and the more sustained response components are highlighted in blue. **f** Labelled RGC from (**e**) (red trace) exemplifies that cells responding with a transient-ON light response have dendrites that stratify in the OFF sublamina (OFF) of the IPL, revealing a native OFF cell anatomy in line with the sign-inversion of optogenetic activation of the Go-TRPM1 signalling axis. **g** Optokinetic reflex measurements revealed significant restoration of OKR acuity in JellyOp-treated *rd1* mice ($p < 0.001$ compared to untreated *rd1* (ANOVA); $n = 9$ WT mice, 6 *rd1* mice, 12 *rd1* JellyOp mice and 15 *rd1* Mela(CTmGluR6) mice; shown is mean ± SD) and a contrast sensitivity (**h**) only slightly worse than Mela(CTmGluR6) treated *rd1* mice ($p = 0.019$ (ANOVA); $n = 7$ WT mice, 6 *rd1* JellyOp mice and 11 *rd1* Mela(CTmGluR6) mice; shown is mean ± SD). Scatter dot plots show mean ± SD. Spike frequency histograms show averages of five traces recorded from the same cell. Light stimuli (500 nm) are shown in green and intensity is indicated in the figure (photons/cm$^2$/s). Scale bars are 50 μm. Source data are provided as a Source Data file.

(8 out of 18 cells, see Supplementary Fig. 5b for other receptive-field types). These transient action potential responses were not present in the RGCs of untreated *rd1* retinas ($n = 16$) and were most likely driven via the night vision circuit (rod-BC to AII amacrine cell to RGC), which feeds rod-OnBC responses into the OFF pathway (Fig. 5d, e)[22]. To test this, we labelled four transient ON RGCs intracellularly with biocytin and confirmed that these cells had dendrites in the OFF sublamina of the retina (Fig. 5f). This apparent inversion of the light response was expected, since the mGluR6 signal is natively driven by glutamate released from photoreceptor cells in the dark while JellyOp, on the other hand, is activated by light[22].

Since visual movement detection relies on rapid retinal signalling, we next probed JellyOp-treated *rd1* mice in a naive optomotor reflex task and determined visual acuity thresholds based on a moving grating stimulus. Figure 5g shows that JellyOp significantly restores the optomotor reflex in blind *rd1* mice compared to untreated littermates ($p < 0.001$). Although the optomotor reflex acuity of JellyOp-treated *rd1* mice was similar to that of Mela(CTmGluR6) treated *rd1* mice ($p = 0.78$; Fig. 5g), the contrast sensitivity was slightly worse ($p = 0.019$; Fig. 5h). These encouraging data, in particular the speed of recovered light responses (Fig. 5d, e), compared to other GPCR-based rhodopsins and rhodopsin chimeras used in vision restoration, suggest that JellyOp may be used as a useful optogenetic tool for Gβγ signalling beyond the restoration of vision[22,25,27–29].

## Discussion

We show that a visual opsin isolated from the box jellyfish, JellyOp, signals via a single constitutively bound Gαs partner. Although some GPCRs do pre-associate with G-proteins, an uninterrupted GPCR/G-protein complex that persists through different GPCR activity states was to the best of our knowledge never described previously[30–32]. JellyOp/Gαs allows us to study, for the first time, a millisecond light-switchable scaled unitary–single GPCR to single G-protein GPCR–response that is independent of G-protein dynamics or cyclic nucleotide exchange (Supplementary Figs. 6,7).

G-protein activation without cyclic nucleotide exchange challenges the current paradigm of GPCR signalling, where nucleotide exchange is considered a prerequisite for G-protein activation. This leads us to suggest an alternative model: (1) In canonical GPCRs, G-proteins are activated through direct interaction with the active GPCR. (2) The activated G-protein, which has a much higher affinity for GTP, rapidly exchanges nucleotides. (3) GTP maintains the active state of the G-protein (Supplementary Fig. 7). In most GPCR signal pathways, the activated G-protein will rapidly inactivate in the absence of GTP. The correlation between nucleotide exchange and a detectable G-protein signal can therefore easily be misinterpreted as a causal link. However, by holding on to its G-protein, JellyOp demonstrates that

nucleotide exchange is most likely an effect rather than a the cause of G-protein activation. In other words, we could think of nucleotide exchanges as a clever mechanism that keeps G-proteins active to amplify signals in a manner that can be dynamically regulated, e.g. through modulators of GTPase activity, rather than a necessary step for G-protein activation.

A lack of G-protein cycling may seem detrimental to signal amplification, and ultimately to light sensitivity. In vertebrate olfactory neurons, a large unitary G-protein response compensates for a low G-protein coupling ratio to allow threshold detection of 19 odour binding events or less[33,34]. A corresponding system in box jellyfish may enable low light vision required for hunting under bioluminescent conditions[35]. Indeed, the lack of G-protein cycling we observe in JellyOp strengthens the thesis that the light sensing cells of box jellyfish and our odour sensing cells share the same evolutionary origin and retained similar advantageous aspects that enhance signal transduction via Gαs[1].

The speed at which JellyOp/Gαs release Gβγ subunits to activate GIRK currents outperforms any known GPCR and advocates its use as an optogenetic tool for rapid control of the Gβγ pathway, independent of AC (Supplementary Fig. 8)[9]. Moreover, we find that a JellyOp-Gαs fusion protein has no apparent coupling to AC (Supplementary Fig. 4). This presents the first optogenetic tool that is selective for the Gβγ pathway. JellyOp is not only fast but also highly efficient in the sense that every JellyOp molecule binds and directly controls the activity of one Gαs partner, subject only to Gαs availability. If JellyOp is over-expressed compared to Gαs, it drives total activation of Gαs. We infer this from the following: (1) The early peak of the JellyOp GIRK response suggests that all activated Gαs molecules are pre-bound (Fig. 1). (2) JellyOp recruits effectively all NES-venus-MiniGαs, which is also over-expressed, to the plasma membrane to form a pre-bound complex (Fig. 2). (3) The JellyOp-Gαs fusion proteins produce significantly larger GIRK currents compared to WT JellyOp and lack the slower "sag" component signalled via Gαi/o (Fig. 1 and Supplementary Fig. 3). In other words, JellyOp seems to have a much lower affinity for Gαi/o subunits and will only signal via this pathway if there are no more Gαs to bind. (4) JellyOp(K72T), which does not pre-bind Gαs and therefore could activate multiple Gαs-proteins in a single activation cycle, does not produce larger GIRK currents (Fig. 4).

A prominent Gβγ signal is a surprising feature for a Gαs-coupled GPCR. Gβγ-activated GIRK channels are normally gated by GPCRs of the Gαi/o class and similarly, TRPM1 channels in OnBCs are gated by Gβγ released from Gαo. JellyOp breaks this rule[6]. One explanation for the low Gαs-to-GIRK coupling observed in other GPCRs is that most activated Gαs subunits do not dissociate entirely from Gβγ and that this leads to less free Gβγ subunits available to activate GIRK channels[8,36,37]. In this frame, an active-state JellyOp/Gαs complex may

hold Gαs in a conformation that catalyses full Gβγ dissociation, either through prolonged, strong or alternative interaction. Concurrently, the high native affinity at which Gαs binds Gβγ permits rapid signal termination once JellyOp/Gαs inactivates.

One common form of GPCR inactivation is mediated through arrestin binding, which should be prohibited by constitutive binding of JellyOp to Gαs, since Gαs occupies the same GPCR binding pocket as arrestin[38]. Surprisingly, a previous study did demonstrate arrestin binding to JellyOp[39]. It is possible that JellyOp binds arrestin at an alternative site on the C-terminus of the opsin. Such distal binding of arrestin, however, was shown to cause GPCR internalisation with uninterrupted G-protein signalling from internalised endosomes[40]. We, however, found that JellyOp-mKate does not internalise after light activation (Supplementary Fig. 9). Nevertheless, it is imperative for any rhodopsin to keep light responses within the dynamic range to prevent saturation at high environmental light intensities. Interestingly, this ties in well with our finding that JellyOp is inactivated by high-intensity violet light. In other words, it is an intriguing prospect that JellyOp could intrinsically adapt to high environmental light intensities by regulating the number of activatable rhodopsin molecules directly through violet light-induced inactivation, for example by accumulation of an inactive syn-photocycle as suggested for channelrhodopsin[41]. The violet component of the solar spectrum is well-suited for light-intensity tuning since its abundance fluctuates dramatically throughout the day, particularly in the shallow-water habitat of box jellyfish, where strong scattering of short wavelengths enhances the relative abundance of violet light at midday[42,43].

We use JellyOp in an AAV-mediated gene therapy to restore Gβγ-mediated light responses in blind mice that suffer from photoreceptor degeneration. JellyOp recovered robust transient light responses, with a response onset that outperformed all G-protein coupled rhodopsins previously used in vision restoration (Fig. 5d)[22,25,27–29]. Microbial rhodopsins, which are currently in clinical trials to restore vision in blind human patients, also drive fast light responses[44]. However, the treatment strategy we use here, which introduce G-protein coupled rhodopsins to the OnBCs, was previously shown to have multiple key advantages[22]. One major advantage is that more signal processing within the retina—even within the OnBCs—is conserved, which potentially translates to a higher quality of recovered vision[25]. A therapy based on JellyOp, or one that shares similar properties, may one day be used to restore visual function in blind people. It would be ironic if the box jellyfish, often dubbed "the world's most poisonous creature", proved to be beneficial to mankind.

## Methods

### Mice
Animal experiments and procedures were in accordance with the Swiss Federal Animal Protection Act and approved by the animal research committee of Bern (approval number BE99/19). Mice were maintained under a standard 12-h light-dark cycle. We used the C3H/HeOuJ retinal degeneration (*rd1*) mouse line as photoreceptor degeneration model, the C57BL/6 J mouse line as wild-type seeing control and the FVB/N Opto-mGluR6-IRES-Turbo635 mouse line for cell sorting of OnBCs[22]. We did not discriminate between male and female mice. Experimental mice were 24 to 60 weeks old.

### Cloning and viral packaging
JellyOp (obtained from Addgene) and human OPN1MW amplified from retinal cDNA was cloned in front of the IRES site of a pIRES2-TurboFP635 vector or in the position of the IRES site of a pIRES2-mKate vector to create fusion proteins[22]. The JellyOp(K72T) mutation was introduced to the pIRES2-JellyOp-TurboFP635 vector using a Phusion mutagenesis kit (ThermoFisher).

For viral production, JellyOp was cloned into a pAAV-770En_454P (h*GRM6*)-JellyOp-IRES2-TurboFP635-WPRE-BGHpA plasmid[20]. Viral

vectors were produced in HEK293 cells by the triple plasmid co-transfection method using the pXX80 helper plasmid and the rep-cap plasmid encoding AAV(7m8)[45] as described in detail elsewhere[46]. The viral titre was $4 \times 10^{13}$ GC/ml. The virus was stored in aliquots at −80 °C until the day of use.

### GIRK whole-cell patch-clamp experiments
A stable HEK293-GIRK1/2 cell line (gift from O. Masseck, RUB), or HL-1 cells, was transiently transfected with opsin constructs using Mirus TransIT®-LT1 transfection reagent (Mirus Bio). After 24 h, whole-cell patch-clamp experiments were performed at 35 °C in a high potassium extracellular solution containing 60 mM KCl, 89 mM NaCl, 1 mM MgCl2, 2 mM CaCl2 and 10 mM HEPES at pH 7.4. In experiments were PKA was blocked, Myr-PKI-14-22 Amide (Merck) was added at a final concentration of 10 μM for 1 h before recording (Supplementary Fig. 8). Patch pipettes had a resistance of ~6 MΩ and were filled with an intracellular solution containing 140 mM KCl, 10 mM HEPES, 3 mM Na₂ATP, 0.2 mM Na₂GTP, 5 mM EGTA and 3 mM MgCl₂, pH 7.4. Cells were voltage clamped at −70 mV while recording GIRK responses to various light stimuli using a HEKA EPC10 amplifier with PatchMaster software. Light stimuli were generated by a pE-4000 system (CoolLED, Andover, United Kingdom) and projected through a 20× water immersion objective onto the recorded cell. The stimulus period was triggered directly by the PatchMaster software. Stimulus intensity was controlled using the pE-4000 system and neutral density filters in the light path. The background light intensity was kept near zero. Traces were analysed offline using Igor Pro software (Wave Metrics). Current amplitudes were normalised to cell size where applicable (pA/pF). The light intensity curve for violet light (Fig. 3e) was generated by fitting a double exponential function to the 405 nm inactivation trace where the time constant of one exponential was specified as the measured dark TauOFF. The time constant of the second exponential, triggered by the violet light stimulus, was then used to calculate the percentage light inactivation after 5 s violet light at the given intensity. Peak spectral sensitivity (Supplementary Fig. 1a) was calculated using a Govardovskii fit[47].

### Bioluminescent plate reader experiments
To visualise changes in Gαs and Gαi/o activity, we transfected HEK293 cells with pcDNA5/FRT/TO Glo22F and with a JellyOp-IRES-TurboFP635 plasmid or with a YFP plasmid (control). Doxycycline was added 5 h after transfection at a concentration of 1 μg/ml. In experiments where Gαi/o was blocked, PTX was also added 5 h after transfection at a concentration of 100 ng/ml. One day after transfection we replaced the culture medium to L15 ($CO_2$-independent with no Phenol red) containing 4 mM beetle luciferin and 1 μM 9-cis retinal (Merck). After allowing 2 h for luciferin uptake, we started luminescence recording using an Infinite F200Pro Tecan plate reader (Männedorf, Switzerland). We activated opsin-based pigments using a custom built LED array and activated β2-adrenergic receptors by adding 25 μl iso-proterenol to each well to a final concentration of 5 μM (Merck). For accurate comparison, data depicted on the same graph were always recorded in parallel on the same 96-well plate. For the cAMP intensity response (Supplementary Fig. 1c), only the first line of wells in a white plastic 96-well plate was illuminated. The resulting drop of light-intensity across neighbouring wells was measured prior to the experiment and used as intensity categories.

### Co-localisation with NES-venus-MiniGαs
HEK293 cells were grown on glass bottom plates in the absence of retinal and transfected transiently with JellyOp-IRES-TurboFP635 and NES-venus-MiniGαs (kind gift form N. Lambert, Augusta University) using Mirus TransIT®-LT1 transfection reagent (Mirus Bio). After 24 h, the culture medium was exchanged to L15-medium (Merck, $CO_2$ independent) and plates were imaged on a Zeiss Laser Scanning

Microscope 880 with Zen (black edition) software. Subsequent experiments were conducted in darkness using only dim far-red illumination. L15-medium containing 2 μM 9-cis retinal (Merck) was added to the cell plates at a 1:1 volume to reach a final 9-cis retinal concentration of 1 μM. The cells were then left in the dark for 10 min before images were taken again. To minimise effects of the imaging light, and capture the "dark state" as accurately as possible, we used a scan speed of 6 and captured the Venus (NES-venus-MiniGαs) channel first. Cells were then exposed to epi-fluorescence illumination (500 nm; $5 \times 10^{13}$ photons/cm$^2$/s; 10 min) before they were imaged again. Changes in fluorescence intensity across the cell membrane were measured in ImageJ v 2.3 (Rasband WS, NIH, Bethesda, Maryland, USA) and normalised in Igor Pro (Wave Metrics).

## AAV transduction
Mice were intravitreally injected at 25–30 weeks of age. For this, they were anaesthetised by intraperitoneal injection of 100 mg/kg ketamine and 10 mg/kg xylazine. The pupil of the right eye was dilated with a drop of 10 mg/ml atropine sulphate (Théa Pharma). We then punctured the dorsal sclera ~1 mm from the corneal limbus using an insulin needle. The insulin needle was removed and a 33 G blunt needle was moved through the pre-made hole to the back of the eye (RPE injection kit from World Precision Instruments). We then injected 2.5 μl of the rAAV vector solution and waited for 2 min before retracting the injection needle form the eye. The second eye was subsequently injected using the same procedure. Following surgery, an antibiotic eye lotion (Isathal from Dechra Veterinary Products) was applied to the eyes to prevent infection and drying of the cornea. Mice were sacrificed 3–5 weeks after injection using isoflurane anaesthesia and cervical dislocation.

## Perforated patch-clamp recordings from isolated bipolar cells
OnBCs were patch-clamped using the perforated, cell-attached method. Isolated cells were prepared by incubating the retina for 45 min at 37 °C in Earle's Balanced Salt Solution supplemented with 40 units/ml papain (lyophilised, Worthington), 5 mM L-cysteine and 0.02% BSA. Papain digestion was followed by gentle titration with a glass pipette before plating cells on Poly-L-Ornithine coated coverslips. Rod-BCs were identified by their characteristic thick axons. Cells were patched-clamped in a recording chamber perfused with Ames medium (Sigma-Aldrich) at 34–36 °C. Patch electrodes were pulled from borosilicate glass to a final resistance of ~10 MΩ. The intracellular solution contained (in mM): KCL 110, NaCl 10, MgCl2 1, EGTA 5, CaCl2 0.5, HEPES 10, GTP 1, cGMP 0.1, ATP 1, and cAMP 0.05. Directly before the experiment, a saturated solution of Amphotericin B in DMSO was added to the intracellular solution at a 1:200 dilution. After adding the Amphotericin B, the solution was vortexed for 1 min and filtered before use. Transfected bipolar cells were identified using a fluorescent reporter (TurboFP635) and targeted for recording under visual control using IR-DIC optics. Light stimuli were generated similar to that described for the HEK-GIRK recordings above. Current recordings were made using a HEKA EPC10 amplifier with PatchMaster software. Traces were analysed offline using Igor Pro software (Wave Metrics).

## FACS sorting and RT-qPCR against Gα subunits
We acutely dissociated 14 retinas from 7 Opto-mGluR6-IRES-Turbo635 mice (p350) as described above. The TurboFP635 expressing OnBCs were isolated on a BD FACSAria™III cell sorter. OnBC RNA was isolated as using a SV Total RNA Isolation Kit (Promega, Dübendorf, Switzerland). We ran one-step quantitative reverse-transcription PCR (qPCR) reactions with 11 ng total RNA using the KAPA SYBR FAST One-Step Universal Kit (Kapa Biosystems, London, UK) on an Eco Real-Time PCR System (Illumina Inc., San Diego, CA). Primers were as follows: GnaS, F 5'- GCCCAGTACTTCCTGGACAA-3', R 5'- TCCACCTGGAACTTGGTCTC-3'; Gnaq, F 5'- ATGACTTGGACCGTGTAGCC-3', R 5'- CCCCTACATCG

ACCATTCTG −3'; GnaO, F 5'- ATGACTTGGACCGTGTAGCC-3', R 5'- CCCCTACATCGACCATTCTG-3'. Values were shown as a fraction of RpL8 mRNA levels, which was normalised to 1.

## Cell-attached patch-clamp recordings from ganglion cells and biocytin injection
The methods for recoding cell-attached light responses from RGCs have been described in detail previously[22]. Electrodes were pulled from borosilicate glass to a final resistance of ~6 MΩ and filled with Ames medium. RGCs were targeted and approached under visual control using IR-DIC optics. Light stimuli were generated similar to that described for the HEK-GIRK recordings above. Voltage recordings were made using a HEKA EPC10 amplifier with PatchMaster software. To label RGCs, we patched cells in the whole-cell configuration using the same intracellular solution described for the OnBCs above but supplemented with 0.2% biocytin (Sigma). The retina was subsequently fixed in 4% paraformaldehyde in 0.1 M phosphate buffer (pH 7.4) for 30 min. Alexa 488 conjugated to streptavidin was used to visualise biocytin-labelled cells (1:400; Invitrogen; S-11223).

## Immunohistochemistry
Immunohistochemistry of cryosections were similar to that described previously[46]. In brief, retinas or eyecups were fixed in 4% paraformaldehyde in 0.1 M phosphate buffer (pH 7.4) for 30 min and cryoprotected in 30% sucrose. Antibodies were diluted in a blocking solution containing 1% Triton-X and 2% donkey serum. Sections were incubated overnight at 4 °C in primary antibody and 2 h in secondary antibody at room temperature. The following antibodies were used: rabbit anti-tRFP (1:1000; Evrogen; AB234) and donkey anti rabbit conjugated to Alexa 488 (1:400; Invitrogen). Nuclei were stained with 10 μg/ml DAPI (Roche). Micrographs were taken on a Zeiss Laser Scanning Microscope 880. Processing of image stacks was done using ImageJ v 2.3 (Rasband WS, NIH, Bethesda, Maryland, USA).

## Optokinetic reflex measurements
Optomotor responses to horizontally drifting, vertically oriented gratings of changing spatial frequency were scored using the Opto-Drum virtual optomotor system (Striatech, v1.2.8) equipped with an automated head-tracking feature. The light intensity at the location of the mouse was set to $4 \times 10^{13}$ photons/cm$^2$/s and the rotation speed kept constant at 12°/s, which was shown to elicit an optimum response under photopic conditions[48]. Experimental data was obtained by two independent and blinded experimenters. All mice were tested in the mornings on 3–4 days over a 3 week period and values averaged.

## Western blot analysis
HEK293 cells were transiently transfected with JellyOp-mKate or JellyOp(K72T)-mKate as described above. Two days after transfection, cells were lysed in standard TNE buffer supplemented with cOmplete™ proteinase inhibitor (Roche). Lysate containing 25 ng protein was incubated in Laemmli buffer for 30 min at 37 °C and run on a 4–20% precast SDS-polyacrylamide gel (Bio-Rad). Protein was transferred to a polyvinylidene difluoride membrane (Immobilion). The membrane was blocked and stained in Tris-buffered saline supplemented with 0.1% Tween 20. The blocking solution (30 min) contained 5% non-fat milk. The primary antibody solution (overnight) contained mouse anti-GAPDH (1:4000, Fitzgerald, 10R-G109A) and rabbit anti-tRFP (1:1000, Evrogen, AB234). The secondary antibody solution (45 min) contained anti-mouse HRP (1:3000, Jackson Immuno Research, 115-035-146) and anti-rabbit HRP (1:3000, Jackson Immuno Research, 111-035-144). Membranes were washed for 10 min between and after incubation steps. Stained membranes were developed using Westar Sun (Cyanagen, XLS063.0250) and imaged on a ChemiDoc MP imaging system (Bio-Rad).

## Statistics

Plots indicate mean ± SD (shaded areas or error bars on graphs). Data in the text equally refer to mean ± SD. The level of significance is illustrated on figure panels as $p \geq 0.05$ (n.s.), $p < 0.05$ (*), $p < 0.01$ (**) and $p < 0.001$ (***). The number of biological samples are illustrated in the figures and in the figure legends. Statistical significance were generally calculated in Microsoft Excel using unpaired Student's *t* tests (one- or two-tailed is specified in the figure legends), except for behavioural data, which was analysed with a one-way ANOVA for multiple comparisons and post hoc analysis using Tukey's honestly significant difference test (HSD) in R v 3.6.0. Assumptions of normality were not rejected by the Shapiro-Wilk normality test and homogeneity of variance was tested with the Levene's or the Bartlett's test. Tau values were obtained from single or double exponential fit functions, where applicable, using Igor Pro 7 (Wave Metrics). Variance always indicate variance between biological replicates except for the RT-qPCR data in Fig. 5a where 14 retinas were pooled and the variance indicated is from technical replicates.

## Reporting summary

Further information on research design is available in the Nature Portfolio Reporting Summary linked to this article.

## Data availability

The source data used in graphs are available in a separate file. The DNA sequences for JellyOp and Mela(CTmGLuRs) are available in GenBank (accession codes AB435549.1 and MQ072285.1, respectively). Source data are provided with this paper.

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

## Acknowledgements

We would like to thank Giulia Schilardi for help with the RT-qPCR experiment, Andrea Madalena for help with the Western Blot experi-ment, Jakub Kralik for contributing to extracellular recordings, Christian Dellenbach for electronic support, Sabine Schneider for support in cloning and viral packaging and Michael Känzig for taking care of the animal facility. Finally, we would like to thank Evi Kostenis, Gregor Cicchetti, David Vaney and Xavier Deupi for constructive comments on the paper. This work was funded by the Swiss National Science Foun-dation (31003A_152807 and 31003A_176065 to S.K.) and the Bertarelli Foundation (Catalyst fund, project BCL7O2 to S.K.).

## Author contributions

M.v.W. collected data, analysed data and contributed to experimental design, data interpretation and writing. S.K. contributed in experimental design, data interpretation and writing.

## Competing interests

The authors declare no competing interests.
