## [Peer Review File · Nature Communications]

A visual opsin from jellyfish enables precise temporal control of G protein signallingEditorial Note: This manuscript has been previously reviewed at another journal that is not operating a transparent peer review scheme. This document only contains reviewer comments and rebuttal letters for versions considered at *Nature Communications*.

REVIEWER COMMENTS

Reviewer #1 (Remarks to the Author):

I think the authors addressed my comments accurately except for one point.

The authors should address one point indicated as follows.

In Fig. S6, the authors suggested an activation mechanism for G proteins that does not require GTP, and they show a schematic presentation including this in Fig. S7. The mechanism of activation of GIRK and AC by Gas/bg protein, including this "GTP-independent activation of G proteins" is of great interest and is an important part of this paper. The authors should add detailed explanation to the schematic presentation (Fig. S7).

For heterotrimeric G proteins, much evidence accumulated over the years established the well-known activation-deactivation mechanism. In the inactive state G proteins exist in trimeric form in which the alpha subunit is bound to GDP. Activated GPCRs facilitate to release the GDP from the alpha subunit and is bound to GTP to dissociate alpha and beta-gamma subunits to produce the activated forms. The alpha subunit has GTPase activity, and therefore GTP is digested to GDP, and the alpha subunit with GDP is bound to the beta-gamma subunit to revert to the inactivated trimeric form. In other words, it is widely accepted that GTP and GDP are required for the activation and inactivation of the G proteins. The authors should describe how the activation/inactivation model of the trimeric G protein involving GTP/GDP is explained in the proposed model for Gs activation by Jellyfish opsin in this paper. I think it is essential to describe about the "central dogma" of this trimeric G protein to explain the GTP (GDP)-free model in details.

Reviewer #3 (Remarks to the Author):

The current changes have improved the MS, and I have no more feedback to give on the novel mechanisms of JellyOp. These are novel and interesting, and I can envision interesting follow-ups, from protein engineering of innovative neurogenetic tools to GPCR evolution. But my judgment is from an outside perspective since GPCRs are not my field.

Regarding the retina results, I appreciate the difficulty of the recordings. However, as presented, they seem not entirely consistent, and I wonder if they obfuscate the interesting characterization of JellyOps dynamics.

1. Because only four recordings are presented, there should be a clear analysis of the all-single traces (all traces for 4 BC should be presented). It is not clear to me if each BC has only one recording or if each BC is also averaged. Moreover, there should be statistical analysis. Even if not significant, the authors can suggest a trend.

2. I see that the authors tried to compare the responses to another Mela(CTmGluR6). Here a statistical analysis between groups should also be done.

3. The spike responses of RGCs are suggested to be mediated through “a disinhibitory circuit that has been well described”. What about classical feed-forward excitation? The depolarization shown in Fig. 5C after the initial hyperpolarization due to JellyOp is stronger and should be observable. I feel the authors have the data, but as presented, it isn't very clear (see also point 6).

4. Moreover, the baseline spiking rate for RGCs in Fig.5D is 20Hz; however, all single traces show no spikes outside the stimulus periods (Fig. S5A). Is this a very selected response type to show repeatability? How many showed repeated responses?

5. If BC recordings are too difficult, RGC voltage clamp recordings would be a solution, and I would encourage them to perform these experiments. This would circumvent the issue with the run-down of BC recordings and the difficulties in perforated patch experiments in BC. Moreover, these experiments are easier to be done and will allow the measurement of several BC inputs at once, increasing SNR, allowing, e.g., the dissection of the feed-forward excitation and the disinhibitory mechanisms and changes in the inputs to repeated stimulation (required for restoration of vision).

6. There is still a knot in my brain regarding the mechanisms of spiking. The authors recorded OFF RGC, that doesn't get (or gets only minimally) direct input from OnBC targeted with JellyOP (by the way, the red arrows are confusing). However, their responses come through indirect disinhibition. JellyOP will first hyperpolarize OnBC, reducing the drive on amacrine cells and disinhibiting OffRGC, making them spike. There are four inversions here. From the 18 blindly recorded RGC, statistically, a few should be ON. After trying to figure this out, I feel that the authors might have the data; one might infer it from a sentence in the figure legend. I would encourage showing the diversity of RGC responses and then making a case for why it would be important to focus on the Off-RGCs. A few sentences clarifying this

are necessary. This is important since one would expect more nuance in the RGC responses, which could be used to make a case. Given that direction selectivity, at least the classical pathway requires direct activation of ON and ON-OFF direction-selective RGCs, that these inputs are necessary for the OKR (Fig. 5G), one expects to have a clear feed-forward excitation too. This would clarify the figure and be a smooth link to Fig. 5G.

7. The behavioral OKR responses are still puzzling me but are clearly striking because they require not just sensing light but also refined retinal computations (see 6). I would be convinced of the link with the voltage clamp experiments of RGC suggested in (5).

I feel that the paper is interesting, particularly because of the characterization and putative use of JellyOp that go beyond the restoration of vision. However, additional work is required to link the results in the retina with the dynamics.

Minor:

The first time I read this paper, I had problems with the OBC acronym since OBC could be On or Off. The same happened the second time I read it. Perhaps changing it to OnBC would make it easier to read.

Abstract: "using a previously unforeseen pathway". Not clear what the pathway is referring to since it sounds like a circuit pathway to me.

Line 39. The colon ":" seems wrongly used, given the structure of the paragraph

Reviewer #4 (Remarks to the Author):

The authors did address all my comments properly.

REVIEWER COMMENTS

Reviewer #1 (Remarks to the Author):

I think the authors addressed my comments accurately except for one point.

The authors should address one point indicated as follows.

In Fig. S6, the authors suggested an activation mechanism for G proteins that does not require GTP, and they show a schematic presentation including this in Fig. S7. The mechanism of activation of GIRK and AC by Gas/bg protein, including this "GTP-independent activation of G proteins" is of great interest and is an important part of this paper. The authors should add detailed explanation to the schematic presentation (Fig. S7).

For heterotrimeric G proteins, much evidence accumulated over the years established the well-known activation-deactivation mechanism. In the inactive state G proteins exist in trimeric form in which the alpha subunit is bound to GDP. Activated GPCRs facilitate to release the GDP from the alpha subunit and is bound to GTP to dissociate alpha and beta-gamma subunits to produce the activated forms. The alpha subunit has GTPase activity, and therefore GTP is digested to GDP, and the alpha subunit with GDP is bound to the beta-gamma subunit to revert to the inactivated trimeric form. In other words, it is widely accepted that GTP and GDP are required for the activation and inactivation of the G proteins. The authors should describe how the activation/inactivation model of the trimeric G protein involving GTP/GDP is explained in the proposed model for Gs activation by Jellyfish opsin in this paper. I think it is essential to describe about the "central dogma" of this trimeric G protein to explain the GTP (GDP)-free model in details.

Thank you for this very interesting commentary. Indeed, JellyOp challenges the current paradigm of GPCR signaling in many ways and GTP-independent signaling is one example. The sequence of events outlined by the referee is most certainly the "standard model" of GPCR signaling. We now propose an alternative model in our discussion (2nd paragraph of discussion) and also in the updated legend of Fig. S7. We agree that it was imperative to do so.

Our proposal of GPCR signaling (in canonical GPCRs): 1) The G-protein is activated through direct interaction with the active GPCR. 2) The activated G-protein has a much higher affinity for GTP compared to GDP and rapidly exchanges nucleotides. 3) GTP maintains the active state.

In most GPCR signal pathways, activated G-proteins will rapidly inactivate in the absence of GTP. This creates a strong correlation between nucleotide exchange and a detectable G-protein signal, which is easily be misinterpreted as a causal link. However, by holding on to its G-protein and keeping it in the active state, JellyOp demonstrates that nucleotide exchange is most likely an effect rather than a the cause of G-protein activation. In other words, we could think of nucleotide exchanges as a clever mechanism that keeps G-proteins active to amplify signals in a manner that can be dynamically regulated, e.g. through modulators of GTPase activity, rather than a necessary step for G-protein activation. A related proposal was made previously (PMID: 15933218).

Reviewer #3 (Remarks to the Author):

The current changes have improved the MS, and I have no more feedback to give on the novel mechanisms of JellyOp. These are novel and interesting, and I can envision interesting follow-ups, from protein engineering of innovative neurogenetic tools to GPCR evolution. But my judgment is from an outside perspective since GPCRs are not my field.

Thank you.

Regarding the retina results, I appreciate the difficulty of the recordings. However, as presented, they seem not entirely consistent, and I wonder if they obfuscate the interesting characterization of JellyOps dynamics.

1. Because only four recordings are presented, there should be a clear analysis of the all-single traces (all traces for 4 BC should be presented). It is not clear to me if each BC has only one recording or if each BC is also averaged. Moreover, there should be statistical analysis. Even if not significant, the authors can suggest a trend.

We now show the 4 individual filtered bipolar cell traces as well as their standard deviation in new Fig. S5D. These are individual traces from different cells and this is now stated in the figure legends.

2. I see that the authors tried to compare the responses to another Mela(CTmGluR6). Here a statistical analysis between groups should also be done.

Since our aim was to demonstrate the speed of JellyOp light responses – as a link to the unique signaling mechanism of JellyOp – we have compared the onset of RGC spiking between JellyOp and Mela(CTmGluR6) treated rd1 retinas. Along the same line, we now compare the timing of the peak hyperpolarization after the onset of the light stimulus in the bipolar cell recordings and demonstrate a highly significant difference between these groups ($p < 0.001$). We thank the referee for this suggestion, which reinforces our findings.

3. The spike responses of RGCs are suggested to be mediated through “a disinhibitory circuit that has been well described”. What about classical feed-forward excitation? The depolarization shown in Fig. 5C after the initial hyperpolarization due to JellyOp is stronger and should be observable. I feel the authors have the data, but as presented, it isn't very clear (see also point 6).

It is apparent from this question and from later questions by the referee that we did not explain our therapeutic approach in a satisfactory manner. We have now done our best to clarify this part of our manuscript.

A common feature of most viral therapies, which target the OnBCs, is that they predominantly target rod-BCs. Our laboratory recently developed short promoters (short enough for use in AAVs), which drive better transfection in cone-OnBCs. Despite this, rod-BCs still make up approximately 80% of all transfected OnBCs in the murine retina, which is rod dominated (PMID: 32258214). The “disinhibitory circuit” that the authors referred to in earlier versions of the manuscript (now changed) was the All-amacrine (rod amacrine) circuit, which feeds the rod-BC signal into both the On and Off cone-BC pathways. It was shown previously that this night vision circuit remains functional in the rd1 retina (e.g. PMID: 25095892).

We now clarify that our OnBC recordings were from rod-BCs. We now also state in the methods, that rod-BCs are easily identified by their relatively thick axons after acute isolation. We show that JellyOp triggers a transient hyperpolarization in rod-BCs and also transient ON responses in some RGCs that have an “OFF” RGC morphology (dendrites in the OFF-sublamina of the IPL). We agree with the referee that the most parsimonious explanation is classic feed-forward excitation from cone-OffBCs to these OFF RGCs, but via rod-BCs that feed onto the All-amacrine circuit, which then feeds onto the cone-OffBCs.

For the second part of the question: Yes, this depolarization can be seen in the spike-frequency-histograms of some RGCs. To demonstrate this point, we added the “blue shaded” background that connect Figs. 5C&E. The spike-frequency-histogram of the cell shown in black does have a biphasic response (the other had no background activity).

4. Moreover, the baseline spiking rate for RGCs in Fig.5D is 20Hz; however, all single traces show no spikes outside the stimulus periods (Fig. S5A). Is this a very selected response type to show repeatability? How many showed repeated responses?

By definition, all light-responsive cells had reproducible responses to light. This includes the cells used to generate the spike-frequency-histograms shown in Fig. 5E, which the referee refers to. As stated in the figure legend, “spike frequency histograms show averages of five traces recorded from the same cell”.

5. If BC recordings are too difficult, RGC voltage clamp recordings would be a solution, and I would encourage them to perform these experiments. This would circumvent the issue with the run-down of BC recordings and the difficulties in perforated patch experiments in BC. Moreover, these experiments are easier to be done and will allow the measurement of several BC inputs at once, increasing SNR, allowing, e.g., the dissection of the feed-forward excitation and the disinhibitory mechanisms and changes in the inputs to repeated stimulation (required for restoration of vision).

Please see answer to question 3 above. As we have now clarified, our OnBC recordings were from rod-BCs, which provide indirect and diverse input to RGCs (via the All-circuit). We now demonstrate diverse receptive-field types in the RGC population of JellyOp treated rd1 retinas (answer to question 6 below; new Fig. S5B). We focused on the transient ON cells because these fast-onset cells is a characteristic feature in JellyOp treated retinas and it links our retinal physiology to the unique coupling mechanism of JellyOp. Also, as we have now clarified, we never intend to claim that RGCs are activated by direct disinhibition of the RGCs themselves.

6. There is still a knot in my brain regarding the mechanisms of spiking. The authors recorded OFF RGC, that doesn't get (or gets only minimally) direct input from OnBC targeted with JellyOP (by the way, the red arrows are confusing). However, their responses come through indirect disinhibition. JellyOP will first hyperpolarize OnBC, reducing the drive on amacrine cells and disinhibiting OffRGC, making them spike. There are four inversions here. From the 18 blindly recorded RGC, statistically, a few should be ON. After trying to figure this out, I feel that the authors might have the data; one might infer it from a sentence in the figure legend. I would encourage showing the diversity of RGC responses and then making a case for why it would be important to focus on the Off-RGCs. A few sentences clarifying this are necessary. This is important since one would expect more nuance in the RGC responses, which could be used to make a case. Given that direction selectivity, at least the classical pathway requires direct activation of ON and ON-OFF direction-selective RGCs, that these inputs are necessary for the OKR (Fig. 5G), one expects to have a clear feed-forward excitation too. This would clarify the figure and be a smooth link to Fig. 5G.

Please see explanations to questions above. The referee is correct in assuming that we should see more receptive-field types in our RGC recordings, we do. We have now made a new supplementary figure to demonstrate other basic receptive-field types in the JellyOp treated rd1 retina (Fig. S5B). We now realize that our focus on fast-onset responses, which links our retinal recordings to JellyOp's unique coupling mechanism, was easily misinterpreted as "the only" RGC response. We thank the reviewer and our updates now aim to rectify this issue.

The main advantage of optogenetic therapies that target OnBCs is that these therapies drive the rod pathway, which feed into all the natural signal processing pathways of the inner retina (subject to "rewiring" in the degenerated rd1 retina). This was the main topic of one of our recent publications (PMID: 36266533).

We have now removed the red arrows from Fig.5.

7. The behavioral OKR responses are still puzzling me but are clearly striking because they require not just sensing light but also refined retinal computations (see 6). I would be convinced of the link with the voltage clamp experiments of RGC suggested in (5).

Please see explanations to questions above.

I feel that the paper is interesting, particularly because of the characterization and putative use of JellyOp that go beyond the restoration of vision. However, additional work is required to link the results in the retina with the dynamics.

Thank you. Constructive comments from the referees certainly helped to clarify and strengthen this link.

Minor:

The first time I read this paper, I had problems with the OBC acronym since OBC could be On or Off. The same happened the second time I read it. Perhaps changing it to OnBC would make it easier to read.

This was not changed.

Abstract: "using a previously unforeseen pathway". Not clear what the pathway is referring to since it sounds like a circuit pathway to me.

This was now changed to "using the previously unforeseen G α s signaling pathway".

Line 39. The colon ":" seems wrongly used, given the structure of the paragraph

This was now changed.

Reviewer #4 (Remarks to the Author):

The authors did address all my comments properly.

Thank you.

REVIEWERS' COMMENTS

Reviewer #1 (Remarks to the Author):

The authors addressed my comments properly. I have no further comment or suggestion on this manuscript.

Reviewer #3 (Remarks to the Author):

The authors addressed my comments and clarified my open questions properly. I have nothing else to add.

Reviewer #1 (Remarks to the Author):

The authors addressed my comments properly. I have no further comment or suggestion on this manuscript.

Thank you.

Reviewer #3 (Remarks to the Author):

The authors addressed my comments and clarified my open questions properly. I have nothing else to add.

Thank you.